# Raman Study of Novel Nanostructured WO_3_ Thin Films Grown by Spray Deposition

**DOI:** 10.3390/nano14141227

**Published:** 2024-07-19

**Authors:** Andreea Gabriela Marina Popescu, Ioan Valentin Tudose, Cosmin Romanitan, Marian Popescu, Marina Manica, Paul Schiopu, Marian Vladescu, Mirela Petruta Suchea, Cristina Pachiu

**Affiliations:** 1National Institute for Research and Development in Microtechnologies—IMT Bucharest, 126A, Erou Iancu Nicolae Street, 077190 Voluntari, Romania; popescu96andreea@gmail.com (A.G.M.P.); cosmin.romanitan@imt.ro (C.R.); fluidproiect@gmail.com (M.P.); marina.manica@imt.ro (M.M.); 2Doctoral School of Electronics, Telecommunications and Information Technology, National University of Science and Technology POLITEHNICA Bucharest, 061071 Bucharest, Romania; schiopu.paul@yahoo.com (P.S.); marian.vladescu@upb.ro (M.V.); 3Center of Materials Technology and Photonics, School of Engineering, Hellenic Mediterranean University, 71410 Heraklion, Greece; tudose_valentin@yahoo.com; 4Chemistry Department, University of Crete, 70013 Heraklion, Greece

**Keywords:** Raman spectroscopy, tungsten oxide, spray pyrolysis, thin films

## Abstract

The present communication reports on the effect of the sprayed solution volume variation (as a thickness variation element) on the detailed Raman spectroscopy for WO_3_ thin films with different thicknesses grown from precursor solutions with two different concentrations. Walls-like structured monoclinic WO_3_ thin films were obtained by the spray deposition method for further integration in gas sensors. A detailed analysis of the two series of samples shows that the increase in thickness strongly affects the films’ morphology, while their crystalline structure is only slightly affected. The Raman analysis contributes to refining the structural feature clarifications. It was observed that, for 0.05 M precursor concentration series, thinner films (lower volume) show less intense peaks, indicating more defects and lower crystallinity, while thicker films (higher volume) exhibit sharper and more intense peaks, suggesting improved crystallinity and structural order. For higher precursor concentration 0.1 M series, films at higher precursor concentrations show overall more intense and sharper peaks across all thicknesses, indicating higher crystallinity and fewer defects. Differences in peak intensity and presence reflect variations in film morphology and structural properties due to increased precursor concentration. Further studies are ongoing.

## 1. Introduction

The response and performance of sensor devices are critically influenced by several factors, including the size, shape, and surface characteristics of the active oxide material. These attributes play a crucial role in defining the electronic and optical properties of oxides, which are known to vary significantly with spatial dimensions and composition [1,2,3,4]. Among the various forms that these materials can take, thin films have emerged as one of the most effective configurations for achieving high sensitivity in sensors [2,3,4,5,6,7,8,9]. This heightened sensitivity is primarily due to the increased number of surface atoms present in thin films, which enhances their interaction with the surrounding environment. The physical and chemical properties of the nano-structured sensing layer, such as tungsten oxide (WO_3_), are notably different from those of their bulk counterparts [2,3,4,5,6]. This disparity arises because the large surface area to volume ratio in nanostructures intensifies surface effects. For sensing applications, it is very important to achieve a high surface-to-volume ratio surface together with a proper thin film structure. In nano-scale dimensions, the effective van der Waals forces, Coulombic interactions, and inter-atomic couplings are significantly altered compared to the bulk material. These modified interactions lead to changes in the electronic structure and chemical reactivity of the material. Specifically, the high surface area in thin films means that a larger proportion of the atoms are exposed to the environment, which can enhance the sensitivity of the sensor [1,2,3,4,5,6,7,8,9,10]. The increased surface interactions enable the material to respond more readily to external stimuli, such as changes in gas concentration, temperature, or light [6,7,8,9,10]. Furthermore, the reduced dimensionality in thin films can result in quantum confinement effects, which can further modify the electronic properties of the material, leading to improved performance in applications like gas sensing, photodetectors, and other optoelectronic devices. The ability to tailor the properties of the oxide material by manipulating its size, shape, and surface characteristics opens new possibilities for designing sensors with enhanced performance [2,3,4,5,6,7,8]. For example, by optimizing the thickness of WO_3_ thin films and controlling their nanostructure, it is possible to achieve a balance between high sensitivity and stability, which is crucial for practical sensor applications. The interplay between surface atoms and the altered inter-atomic forces in these nanostructured films provides a rich field of study for developing advanced materials with superior sensing capabilities [2,3,4,5,6,7,8,9,10]. The performance of sensor devices is intricately linked to the characteristics of the active oxide material, particularly in its nano-structured thin film form [6,7,8,9,10]. The unique properties that arise from the high surface area and modified inter-atomic interactions in these materials pave the way for highly sensitive and efficient sensors, making thin films a preferred choice in the design of next-generation sensing technologies. 

WO_3_ is a typical n-type wide bandgap semiconductor like most transition metal oxides. As a gas-sensing material, it has attracted considerable attention, and it is used to detect various hazardous gases [2,3,4,5,6,7,8,9,10,11]. This is only one of its applications, among others, such as photochromic [12] electrochromic [13,14], and gasochromic “smart windows” [15,16,17]. Previous studies indicated that its unique nanostructuring during growth is highly related to the sensing performances. Countless advances have been achieved to design and fabricate diverse WO_3_ in different forms. The methods used for WO_3_ deposition, such as PVD (thermal evaporation [18,19], DC- [20], RF- [19] and magnetron sputtering [21]) and CVD [22], and sol-gel [19] usually lead to polycrystalline or porous thin films with a homogeneous distribution of grains onto the substrate surface. The use of nonvacuum methods for deposition offers significant advantages, including cost-effectiveness, as they eliminate the need for expensive vacuum equipment and maintenance. These methods enable large-area and flexible substrate coating, enhancing scalability for industrial applications [2,3]. Nonvacuum techniques are more environmentally friendly, reducing energy consumption and hazardous waste. They also allow for simpler, more versatile processing, accommodating a wide range of materials. Additionally, nonvacuum methods often provide better control over film properties and can be integrated with existing manufacturing processes [3,4,5,6]. In the particular case of this study, using the spray deposition method followed by thermal annealing, except for the cheap and cost-effective large-area deposition, offers the possibility of achieving unique 3D structuring of the surface and elevated surface-to-volume ratio values. However, it is still challenging to achieve high performance. To significantly enhance the gas-sensing properties, many strategies have been explored. Although the mechanisms responsible for WO_3_ gas sensing are not completely understood, an empirical model to explain the fundamental gas sensing mechanism is widely accepted and is as follows. Upon exposure to air, oxygen molecules will be chemisorbed on the WO_3_ surface, inducing the creation of an electron depletion layer by attracting electrons from the conduction band. The adsorbed oxygen can further evolve into active oxygen species such as O^−^, and O_3_^−^ at a certain temperature [11,23,24], which is highly reactive with target gas molecules. The charge carriers are affected by the presence of target gas. For example, a reduction reaction can occur between reducing gases and these oxygen species, leading to the release of the trapped electrons back. As a result, the change in resistance reflects the concentrations of the target gas. In the past years, typical 2D WO_3_ structures such as thin films have drawn considerable attention in the gas-sensing field due to the high surface-to-volume ratio, modulated surface activities, surface polarization, and rich oxygen vacancies. Many studies on WO_3_ used as sensing layers are present in the scientific literature, but this material is still challenging [6,7,8,9,25,26]. Raman spectroscopy can detect different stoichiometries and structures, including polymorphs that contain the same atoms but in different crystalline forms [26]. This technique allows for the analysis of mixtures and the quantification of their chemical composition. Despite the extensive research on WO_3_ thin films, a Raman spectroscopy study examining the effects of thickness variations correlated with the initial concentration of the precursor has, to our knowledge, not been performed before. This specific investigation is important, as it could provide new insights into how these variables influence the material’s structural and chemical properties, which are critical for optimizing its performance in various applications. By leveraging Raman spectroscopy, one can detect subtle changes in the WO_3_ thin films’ structure and composition induced by varying the thickness and precursor concentration. This method’s sensitivity to different stoichiometries and crystalline forms makes it particularly suited for such an analysis. Therefore, conducting this study could potentially uncover novel information about the interplay between film thickness, precursor concentration, and the resultant properties of WO_3_ thin films. This could lead to improved methodologies for fabricating thin films with tailored properties for specific technological applications. The scope of the present study is to observe and correlate structural properties characterized by Raman spectroscopy for two kinds of pure WO_3_ nanostructured thin film thickness series grown from precursors with two different concentrations, leading to a rare structuring of the surface, which was only briefly noticed in the scientific literature. As mentioned above, up to our knowledge, a Raman spectroscopy study of the thickness variations effect correlated with an initial concentration of the precursor has never been performed before on WO_3_ thin films. This is a specific niche in materials science research. While WO_3_ thin films and their properties have been extensively studied, the unique combination of variables—thickness variation and initial precursor concentration, specifically analyzed through Raman spectroscopy is novel.

## 2. Materials and Methods

WO_3_ thin films’ deposition was performed using custom-made air carrier flow static spray pyrolysis equipment. The films’ growth was carried out at a vertical position and at a 30 cm distance between the spray nozzle and the substrate, which was kept at a temperature of 250 °C during deposition. The precursor solution was prepared by dissolving the required amount of tungstic acid H_2_WO_4_ in distilled water to obtain 0.1 M and 0.05 M solutions. Three different quantities (5, 8, and 12 mL) were employed since the thickness of the films can be controlled by the volume of the solution. Thickness could not be measured to an accurate level due to the complex surface morphology of the films. The growth time increased with increase in the volume of solution, while the flow of the air carrier remained constant to ensure a constant deposition rate. Tungstic acid (H_2_O_4_W), ≥99.0% (calcined substance, T) powder was purchased from Sigma-Aldrich (Sigma-Aldrich, St. Louis, MO, USA) and distilled water was used as a solvent for the precursor’s preparation. Fluorine-doped Tin Oxide (FTO—F:SnO_2_) coated glass was the substrate. After the spray deposition of the thin films, thermal annealing was carried out for 2 h at 450 °C in oxygen atmosphere to increase films’ adhesion to the substrate and improve crystallinity. 

X-ray diffraction (XRD, Rigaku, Tokyo, Japan) was performed using a Rigaku ultra high-resolution triple-axis multiple reflection SmartLab X-ray Diffraction System, Tokyo, Japan. Scanning electron microscopy (SEM) characterization was done using a field emission Nova NanoSEM 630 (FEI Company, Hillsborough, OR, USA) and a field emission scanning electron microscope (FE-SEM) without sample preparation. Detailed Raman spectroscopy analysis was carried out using a Witec Raman spectrometer (Alpha-SNOM 300 S, WiTec GmbH, Ulm, Germany). Raman spectroscopy was performed to observe and correlate phase purity and structural properties of the WO_3_ films under a visible (532 nm) laser excitation using the Witec Raman spectrometer equipped with a confocal microscope (Olympus 100×, Tokyo, Japan) and the visible excitation generated by a Nd-YAG doubled diode pumped laser with output power of 50 mW. The Raman spectra of the tungsten oxide were taken under ambient conditions with the spectral acquisition time 20 s/scan.

## 3. Results and Discussions

SEM imaging at low magnification was conducted on six samples selected upon the precursor’s volumes and concentrations: 5, 8, and 12 mL at 0.1 M and 0.05 M. The SE images displayed in Figure 1 disclose significant morphological variations on large areas of the WO_3_ layer, as seen below:

It can be observed that for a specific precursor concentration, the increasing thickness leads to a surface morphology with increased number of features. One can observe the presence of small islands alternating with wall-like structures, with size increasing as the thickness increases. With respect to the precursor concentration, higher molar concentration for the precursor solution leads to similar surface morphology at small thickness, but as the thickness increases, the surface morphologies change, and at lower precursor concentration, a higher number of wall-like structures appear, while the island-like structures’ number and size decreases. At higher precursor concentration, the surface morphology seems to evolve towards larger island-like structures as the thickness is increasing.

Grazing incidence X-ray diffraction was employed to study the constituent phases of WO_3_ films. For this purpose, the incidence angle was kept at 0.5°, while the detector scanned in the 2θ range of 20–60°, as shown in Figure 2.

The XRD data reveal the presence of single-phase WO_3_. Each diffraction pattern presents a set of diffraction features with different intensities located at 23.1, 23.5, 24.2, 26.5, 28.7, 33.3, 34.1, 35.4, 41.7, 44.4, 45.6, 47.2, 48.3, 49.9, 50.7, and 55.8°, assigned with different (hkl) Miller indices using ICDD database—International Center for Diffraction Data. The main diffraction peaks were labeled with the corresponding set of Miller indices. These were attributed to WO_3_ with monoclinic crystal structure with a preferred orientation of the crystallites along the c-axis, belonging to P21/n(14) space group with the following lattice parameters and angles: a = 0.73 nm; b = 0.75 nm; c = 0.76 nm; α = 90°; β = 90.9°; γ = 90°, according to card no. 083-0951. No impurities’ peaks were detected in the XRD patterns. A monoclinic phase was also observed by different authors in previous investigations on WO_3_ [6,19,20,21,22,23,24]. For instance, Dongale and co-workers [24] reported monoclinic WO_3_ films obtained using spray pyrolysis at different substrate temperatures. Acosta et al. [24] reported monoclinic phase of WO_3_ with a preferred orientation of the crystallites along the c-axis that corresponds to surface texturing, while Ortega et al. [27] showed the hexagonal and monoclinic phase co-existence on glass substrate and a pure monoclinic phase on FTO for WO_3_ deposited by pulsed spray pyrolysis. Different deposition parameters do not affect the position of the diffraction peaks, which indicates that the interplanar distance of the monoclinic-WO_3_ remains constant along different precursor volumes and concentrations. However, the fitting of the main diffraction peaks located at 24.3°, with a pseudo-Voigt function indicates that the crystal quality is affected by different deposition parameters. The value of the size of the crystalline domains (also called mean crystallite size), *τ* can be obtained from the Full Width at Half Maximum (FWHM) of the diffraction peak *β* in the following way [16,28]: (1)$$\tau =\frac{k\lambda }{\beta \cos\theta }$$

Here, *k* = 0.93 is the shape factor of the crystallites, and *λ* = 0.15406 nm is the X-ray wavelength provided by the monochromatic X-ray CuKα1 source. 

According to the pseudo-Voigt fit, at 0.05 M for the lowest precursor’s volume (5 mL), *β* = 0.51°. Further, when the precursor’s volume increases, *β* decreases to 0.46° (8 mL), reaching 0.45° (12 mL). Similarly, at 0.1 M, *β* decreases from 0.51° (5 mL) to 0.43° (8 and 12 mL). According to Equation (1), when increasing the precursor’s volume, the crystal quality becomes better, which is reflected by the mean crystallite size increasing from ~16 nm to ~19.5 nm. The obtained values for the mean crystallite size were compared with Rietveld refinement and Willamson-Hall plot results, showing comparable values. An increase in the precursor’s volume seems to favor an increase in the crystallite size. These observations are also supported by the SEM micrographs, which pointed out that at higher precursor concentrations, the surface morphology evolves towards larger island-like structures. Further, one can observe that the different deposition parameters are related to different Raman signatures. 

Raman spectroscopy results for the WO_3_ thin films grown at precursor’s volumes and concentrations: 5, 8, and 12 mL at 0.1 M and 0.05 M are presented in Figure 3a,b.

As already mentioned, Raman spectroscopy offers important information about the crystalline phases of WO_3_ crystal. Tungsten trioxide thin films can exist in different phases depending on synthesis temperature, i.e., (a) low-temperature phase (m-WO_3_)—this is the most stable phase of WO_3_ at ambient conditions corresponding to the monoclinic crystal structure. The α-phase can be synthesized at relatively low temperatures, typically below 740 °C [29]; (b) high-temperature phase (at higher temperatures, typically above 740 °C, α-WO_3_ transforms into the c-phase a cubic (ideal) crystal structure; (c) the phase transitions in the WO_3_ thin films occur in sequence as the temperature is increased: monoclinic (m-WO_3_) → orthorhombic (β-WO_3_) → tetragonal phase (α-WO_3_) [30].

For sensing functions, the WO_3_ needs to be nanostructured and have a large surface area to enable the analytes to diffuse through the film. Acentric nature and spontaneous electric dipole moment of ferroelectric ε–WO_3_, for example, leads to increased interaction with high dipole moment analytes such as acetone [31], which is used for medical devices sensing the acetone level in human breath in concentrations of parts per billion (ppb) for non–invasive blood glucose monitoring [32]. In the monoclinic phase, photoelectrochemical and photocatalytic properties are enhanced when the film is highly crystalline and preferentially oriented because this highly crystalline structure will have fewer defects when acting as the recombination center and should suppress mutual e––h+ recombination [33]. According to some authors, polycrystalline WO_3_ film has almost no photochromic sensitivity, whereas amorphous WO_3_ has high photochromic and electrochromic sensitivity due to high surface area [34]. 

The fundamental Raman vibrational modes within the lattice of WO_3_ contain stretching (ν), bending (δ), and out-of-plane wagging (γ) modes.

In the monoclinic m-WO_3_ crystalline structure, these vibration modes are observed at ≅803, 714, and 270 cm^−1^, corresponding to the stretching of ν (O-W-O) bonds, stretching of ν (W-O) bonds, and bending of δ (O-W-O) bonds, respectively [35].

Comparing the Raman spectra (Figure 3) obtained for WO_3_ films made by spray deposition method with results reported in the literature using other deposition techniques (Chemical Vapor Deposition [36], Chemical Solution Deposition [37,38], or Dip-Coating [39]), one can observe a dominant presence of vibration modes at 270 cm^−1^, 714 cm^−1^, and 805 cm^−1^. This indicates a monoclinic structure of m-WO_3_ but with changes in the Raman spectrum of the resulting material induced by the deposition method. 

The O-W-O and W-O bonds in monoclinic structure remain unchanged, with variations only in intensity with increases in volume for spray-coating (for 0.5 M δ (O-W-O), I 12 mL/I 5 mL = 1.96), and they are less sensitive to higher molar masses (for 0.1 M δ (O-W-O), I 12 mL/I 5 mL = 0.15). 

The WO_3_ Raman spectra (Figure 3a—0.05 M probes) display bands at low wavenumbers, observed within the wavenumber range of 80–400 cm^−1^ corresponding to the lattice modes. Above all, for Figure 3b—0.1 M probes with band identification, there is a cluster of bands at 89, 131, 190, and 266 cm^−1^. The bands below 270 cm^−1^ are attributed to low-frequency phonon charge markers. The band at 266 cm^−1^ is attributed to lattice modes’ ν (O-W-O) vibration; the other bands correspond to W-W lattice modes’ vibration. At the same time, the Raman bands corresponding to the W-W bonding located in the 200 cm^−1^ region become more intense, being closely related to the increased structural order in the spray-deposited WO_3_ material (Figure 4, 12 mL volume sprayed). The bands between 270 and 323 cm^−1^ are the typical modes indicating the crystalline quality of the WO_3_ films [26]. The high Raman peak at 266 cm^−1^ indicates that the structure contains an important fraction of the crystalline phase [38]. 

In this case, the deposition process influenced the crystalline structure of WO_3_, altering the vibration mode of certain atomic bonds in the material, such as a weak δ (O-W-O), reflected through weak vibration modes in the Raman spectrum (Figure 3), as well as the weak presence of O-lattice bonds in the spectral region around 600 cm^−1^ (absence of predominant O-lattice modes at frequencies of approx. 604 and 674 cm^−1^—Table 1). All these changes are induced by deviations from the ideal stoichiometry of the WO_3_ composition, resulting in a variation of the atomic ratio between tungsten and oxygen as confirmed by EDX studies results not included in this communication. 

The bond at 714 cm^−1^, also known as the γ (W-O) band, in the Raman spectrum of WO_3_, indicates the vibrations of tungsten atoms bonded to oxygen atoms in the crystalline lattice. The disappearance of this band may be associated with the loss of some oxygen atom bonds in the crystal lattice. 

The band at 806 cm^−1^ represents the Raman vibration mode of crystalline WO_3_ (m-phase), indicating the stretching vibrations of the bridging oxygen, ν_a_ (O-W-O). 

The disappearance of symmetric ν_s_ (O-W-O) may be caused by chemical reactions, leading to the modification or destruction of terminal oxygen-containing groups such as OH- and COO- in the WO_3_ lattice. This could result from interactions with ambiental compounds during the deposition process.

The deposition of WO_3_ by spray-coating induces slight shifts in the frequency of the Raman bands (Figure 4c γ (W-O)). These shifts may be caused by changes in the local environment of the tungsten atoms or alterations in the interactions between atoms and light.

Another consequence of using the spray deposition process seems to be the decrease in FWHM Raman modes: FWHM for ν (O-W-O) mode for 5 mL decreased by ≈12%, and the same behavior is observed for the modes γ (W-O) and ν_a_ (O-W-O) as well. 

All these changes in the Raman spectrum of the spray-deposited material may be associated with the improved crystallization of the material and a reduction in the dispersion of vibration modes in the WO_3_ crystalline structure. The detailed Raman spectroscopy analysis of spray-deposited WO_3_ films provides a comprehensive understanding of structural characteristics, crystalline properties, and the influence of deposition parameters on material quality. These insights are crucial for optimizing WO_3_-based sensors and other applications requiring precise control over material structure and properties. Summarizing, the influence of thickness and precursor concentration

0.05 M Series: Thinner films (lower volume) show less intense peaks, indicating more defects and lower crystallinity. Thicker films (higher volume) exhibit sharper and more intense peaks, suggesting improved crystallinity and structural order.0.1 M Series: Films at higher precursor concentrations show overall more intense and sharper peaks across all thicknesses, indicating higher crystallinity and fewer defects. Differences in peak intensity and presence reflect variations in film morphology and structural properties due to increased precursor concentration.

## 4. Conclusions

Using the spray deposition method, wall-like structured monoclinic WO_3_ thin films were obtained by for further integration in gas sensors. As for sensing applications, it is very important to achieve a high surface-to-volume ratio surface together with a proper thin films structure and stoichiometry, the WO_3_ thin films developed during this work seems to offer excellent premises for use as sensing layers. The effect of the sprayed solution volume variation (as a thickness variation element) on the detailed Raman spectroscopy for WO_3_ thin films with different thicknesses grown from precursor solutions with two different concentrations was analyzed. A detailed analysis of the two series of samples showed that the increase in thickness strongly affects the films’ morphology while their crystalline structure is only slightly affected. Bands at low wavenumbers (e.g., 89.73 cm^−1^, 110.83 cm^−1^, 180 cm^−1^) are associated with lattice vibrations and defect states, while higher wavenumber bands (e.g., 674 cm^−1^, 714.96 cm^−1^, 803.91 cm^−1^) correspond to stretching vibrations indicative of crystallinity. A higher precursor concentration (0.1 M) and increased film thickness enhance crystallinity, as evidenced by the sharper and more intense Raman peaks. This analysis provides a comprehensive understanding of how synthesis parameters affect the vibrational modes and overall quality of WO_3_ thin films. It was observed that, for both concentrations, the increase in thickness seemed to slightly affect the stoichiometry of the material and, in association with surface morphology variations, provide new possibilities for the fine-tuning of films’ properties towards enhanced sensing applications. Further studies are ongoing.

## Figures and Tables

**Figure 1 nanomaterials-14-01227-f001:**
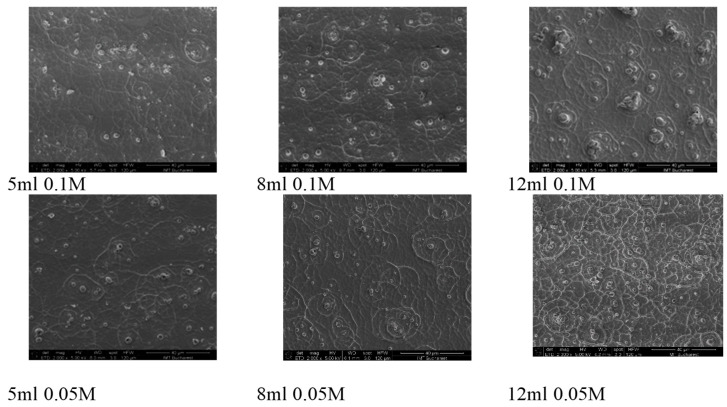
SEM images of WO_3_ thin films grown at precursor volumes and concentrations: 5, 8, and 12 mL at 0.1 M and 0.05 M (scale 40 µm).

**Figure 2 nanomaterials-14-01227-f002:**
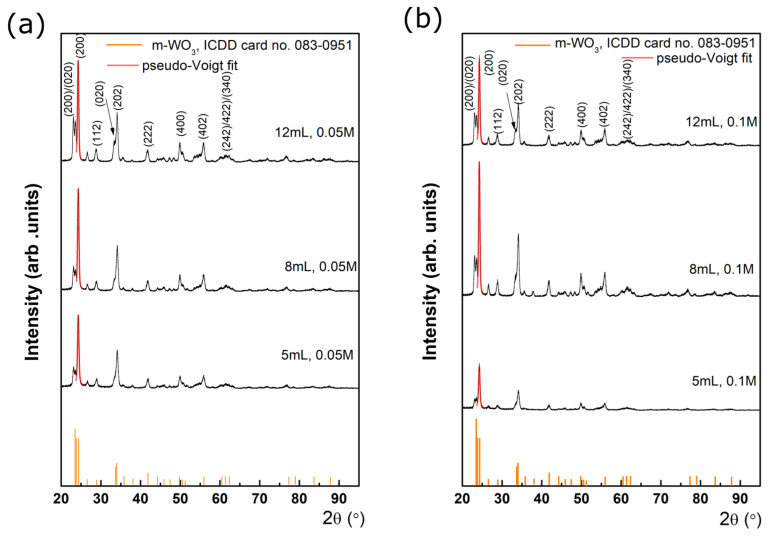
XRD patterns of WO_3_ thin films grown at precursor’s volumes and concentrations: 5, 8, 12 mL at (**a**) 0.05 M and (**b**) 0.1 M.

**Figure 3 nanomaterials-14-01227-f003:**
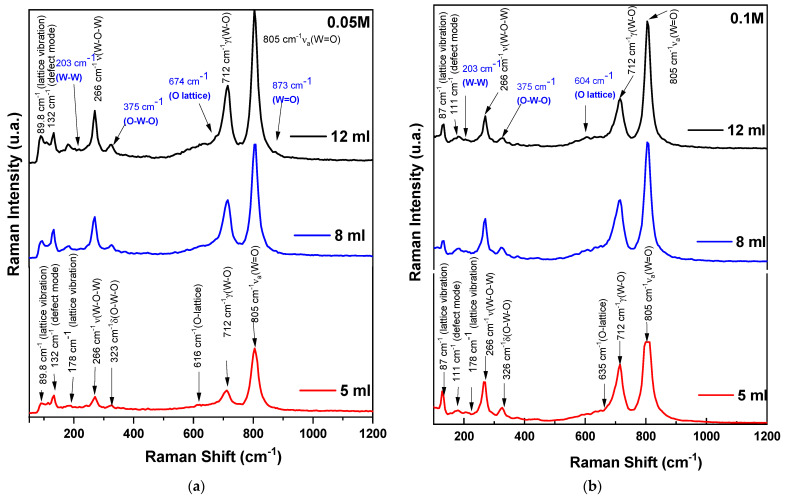
Raman spectroscopy results for WO_3_ thin films grown at precursor’s volumes and concentrations: 5, 8, 12 mL at (**a**) 0.05 M and (**b**) 0.1 M.

**Figure 4 nanomaterials-14-01227-f004:**
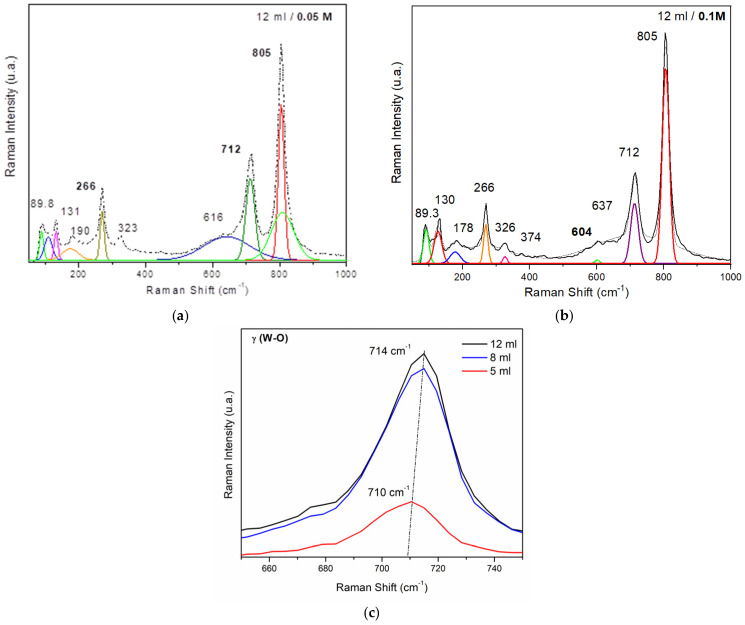
(**a**,**b**) Examples of the deconvoluted Raman spectra of the 12 mL WO_3_ at the two different concentrations; (**c**) Raman intensity variation with Raman shift for the films with different thicknesses.

**Table 1 nanomaterials-14-01227-t001:** Summary of WO_3_ Raman bands.

Raman Band (cm^−1^)	Raman Assignment
89.73	Low-frequency phonon change marker
	Low-energy lattice vibrations. **Discussion**: This band is typically associated with external lattice vibrations. It is prominent in the 0.05 M series, indicating structural ordering with potential lattice imperfections. Its absence or lower intensity in the 0.1 M series suggests higher crystallinity and fewer defects.
110.83	Defect-related or localized vibrational modes. **Discussion**: This band may indicate specific defect states or unique structural features. It is less commonly reported and may appear as a weak peak, reflecting lattice imperfections or localized modes due to impurities.
180	Low-frequency phonon change marker
	Lattice vibrations or phonon modes. **Discussion**: Present in both 0.05 M and 0.1 M series, indicating a stable characteristic of the WO_3_ lattice. Its consistent presence suggests that this mode is relatively unaffected by precursor concentration but varies in intensity with film thickness.
203.89	W-W
	Lattice vibrations. **Discussion**: This band is indicative of specific W-W lattice vibrational modes, observed prominently in the 0.05 M series. It reflects the differences in film morphology and thickness.
270.02	ν (O-W-O) in monoclinic phase
	Bending modes. **Discussion**: Associated with W-O-W bending vibrations. Its presence is related to monoclinic phase structural characteristics specific to the WO_3_ thin films.
326.31	δ (O-W-O)
	Out-of-plane wagging modes. **Discussion**: Indicates out-of-plane bending or wagging of W-O bonds. Prominent in both concentration series, reflecting the structural integrity of the WO_3_ lattice.
374	δ (O-W-O)
	Lattice vibrations. **Discussion**: This band, observed in the 0.1 M series, indicates complex lattice dynamics and interactions. Its presence suggests differences in structural properties influenced by higher precursor concentration.
604	O-lattice
	Bending modes. **Discussion**: This band is related to O-W-O bending vibrations. More prominent in the 0.1 M series, indicating a more ordered structure due to higher precursor concentration.
674	O-lattice
	W=O stretching vibrations. **Discussion**: Indicates strong W=O stretching, present in the 0.05 M series. It suggests moderate crystallinity and possible defects.
714.96	γ (W-O)
	W=O stretching vibrations. **Discussion**: Common in both concentration series, indicating strong W=O stretching vibrations. Higher intensity in the 0.1 M series reflects better crystallinity.
803.91	ν_a_ (anti-symmetric O-W-O) monoclinic phase
	W=O stretching vibrations. **Discussion**: Present in both series, indicating strong W=O stretching. The higher intensity in the 0.1 M series suggests improved structural order.
873	ν_s_ (symmetric W=O terminal)
	High-frequency stretching modes. **Discussion**: This band is indicative of high-frequency W=O stretching vibrations, observed prominently in both series. Its intensity reflects the crystallinity and thickness of the films.

## Data Availability

The raw and processed data required to reproduce these findings cannot be shared at this time due to technical or time limitations. The raw and processed data will be provided upon reasonable request until the technical problems have been solved.

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
