# Peer review of "Raman Study of Novel Nanostructured WO3 Thin Films Grown by Spray Deposition"

_nanomaterials, 2024, doi:10.3390/nano14141227_

Round 1

Reviewer 1 Report (Previous Reviewer 2)

Comments and Suggestions for Authors

The manuscript has been considerably improved, however, there are still some issues that should be additionally clarified. Please follow the comments below.

1. Cancel from the Abstract lines 30 and 31 ”For sensing applications, it is very important to achieve a high surface-to-volume ratio surface together with a proper thin films structure. Further studies are ongoing”. These could be added to the Introduction part and Conclusions. respectively.

2. Rewrite this “WO3 is a typical n-type gas-sensing material that attracted considerable attention and it is used to detect various hazardous gases among others such as photochromic [2] electrochromic [3], and gasochromic “smart windows” [4]”. It is an n-type wide bandgap semiconductor like most transition metal oxides. The photochromic, electrochromic, and gasochromic “smart windows” are another type of important applications.

3. Table 1 is very nice, however, add the peaks/bands assignments to the Raman spectra too.

4. Please add that “the monoclinic phase of WO3 with a preferred orientation of the of crystallites along the c-axis” is a textured WO3.

5. The Ref. 4 from your list [Sol. Energy Mater. 2016 Volume 144, pages 316-323] is nice but not the original one on the gasochromic effect demonstration in WO3 and study. You should refer first to the papers of Prof. L.-K. Thomas [https://www.researchgate.net/scientific-contributions/L-K-Thomas-73041700], from TU-Berlin, who had the patent for the gasochromic effect, and then add the new paper [4] dated 2016.

6. In the Introduction part (lines 74 and 75) please make clear that amorphous WO3 films can be grown by RT DC-sputtering [https://doi.org/10.1016/j.jmst.2020.11.036], RT thermal evaporation [..] and low-temperature CVD [DOI: 10.1016/S0022-0248(98)01051-3].

7. It would be nice if you could add some Raman spectra of the as-deposited thin films before annealing.

8. Granqvist has proposed in Ref. [C.G. Granqvist, Appl. Phys. A 57 (1993) 3] in 1993 that ionic transport is greater in a loosely packed material (highly porous amorphous). This is the reason for higher coloration efficiency of amorphous WO3 in electrochromic, gasochromic, etc., effects. Deviation from stoichiometry plays a certain role too [L.-K. Thomas]. Rewrite “According to some authors, polycrystalline WO3 film has almost no photochromic sensitivity whereas amorphous WO3 has high photochromic and electrochromic sensitivity due to high surface area [22] (lines 250-251) adding the Granqvist's results.

9. Show the impact of film thickness on the crystallinity of as-deposited tungsten oxide thin films by different methods (PVD and CVD). Compare their porosity values with your own and underline the cost-effectiveness of your method.

10. Shorten the Conclusions. This is not necessary “The Raman analysis contribute to refining the structural observations by Raman anal XRD. analysis of Raman bands in WO3 thin films reveals significant insights into their structural properties (lines 332- 333).” Only the results achieved should be included.

11. English still could be slightly improved.

Comments on the Quality of English Language

Minor editing of English language required

Author Response

We thank the reviewer for his time and useful suggestions. All the constructive suggestions were carefully considered and accommodated in the improved revised manuscript. Detailed answers to the reviewer’s comments are provided bellow between the lines. We must mention that there are still some formatting issues and perhaps minor English typos that will be solved in collaboration with mdpi editorial team after the manuscript acceptance. We hope that our contribution is now ready to be accepted.

The manuscript has been considerably improved, however, there are still some issues that should be additionally clarified. Please follow the comments below.

  1. Cancel from the Abstract lines 30 and 31 ”For sensing applications, it is very important to achieve a high surface-to-volume ratio surface together with a proper thin films structure. Further studies are ongoing”. These could be added to the Introduction part and Conclusions. respectively.

Done. Thank you for your suggestion.

  1. Rewrite this “WO3 is a typical n-type gas-sensing material that attracted considerable attention and it is used to detect various hazardous gases among others such as photochromic [2] electrochromic [3], and gasochromic “smart windows” [4]”. It is an n-type wide bandgap semiconductor like most transition metal oxides. The photochromic, electrochromic, and gasochromic “smart windows” are another type of important applications.

Done. Thank you for your suggestion

  1. Table 1 is very nice, however, add the peaks/bands assignments to the Raman spectra too.

Done. Thank you for your suggestion. Both, fig 3 and 4 were improved.

  1. Please add that “the monoclinic phase of WO3 with a preferred orientation of the of crystallites along the c-axis” is a textured WO3.

Done. Thank you for your suggestion.

  1. 4 from your list [Sol. Energy Mater. 2016 Volume 144, pages 316-323] is nice but not the original one on the gasochromic effect demonstration in WO3 and study. You should refer first to the papers of Prof. L.-K. Thomas [https://www.researchgate.net/scientific-contributions/L-K-Thomas-73041700], from TU-Berlin, who had the patent for the gasochromic effect, and then add the new paper [4] dated 2016.

With all due respect, since our contribution is a communication focused on a different main subject we try to keep it strictly connected to its main subject… We added the most recent publication of Prof. L.-K. Thomas group that cite and base anyway on the previously published studies. We added also a very recent review on the subject. We hope that the reviewer understand and agrees with us.

  1. In the Introduction part (lines 74 and 75) please make clear that amorphous WO3 films can be grown by RT DC-sputtering [https://doi.org/10.1016/j.jmst.2020.11.036], RT thermal evaporation [..] and low-temperature CVD [DOI: 10.1016/S0022-0248(98)01051-3].

With all due respect, as explained also before, since our contribution is only a communication, we try to avoid overloading it with references that are not strictly connected to the main subject… We hope that the reviewer understands and agrees with us.

  1. It would be nice if you could add some Raman spectra of the as-deposited thin films before annealing.

Based on our previous experimental experience, the thermal treatment is part of the fabrication/growth process and we cannot provide such spectra... Detailed thermal treatment effects studies may be the subject of another work…

  1. Granqvist has proposed in Ref. [C.G. Granqvist, Appl. Phys. A 57 (1993) 3] in 1993 that ionic transport is greater in a loosely packed material (highly porous amorphous). This is the reason for higher coloration efficiency of amorphous WO3 in electrochromic, gasochromic, etc., effects. Deviation from stoichiometry plays a certain role too [L.-K. Thomas]. Rewrite “According to some authors, polycrystalline WO3 film has almost no photochromic sensitivity whereas amorphous WO3 has high photochromic and electrochromic sensitivity due to high surface area [22] (lines 250-251) adding the Granqvist's results.

The reviewer would be right if our work would be focusing on electrochromic properties... Since our contribution is a communication focused on different research subject, we try to keep it connected to the main subject although we added also a very recent review on the subject… We hope that the reviewer understands and agrees with us.

  1. Show the impact of film thickness on the crystallinity of as-deposited tungsten oxide thin films by different methods (PVD and CVD). Compare their porosity values with your own and underline the cost-effectiveness of your method.

With all due respect, since our material structuring is quite different of the other materials previously obtained even by spray pyrolysis and they consists on 3D structures, we consider that comparing our materials with thin films obtained by different methods it is not quite proper in the case of this communication… A relevant comparison we would be able to present in later articles, after the next steps of the present study, after a proper evaluation of the sensing properties. We hope that the reviewer understands and agrees with us.

  1. Shorten the Conclusions. This is not necessary “The Raman analysis contribute to refining the structural observations by Raman anal XRD. analysis of Raman bands in WO3 thin films reveals significant insights into their structural properties (lines 332- 333).” Only the results achieved should be included.

Done. Thank you for your suggestion.

  1. English still could be slightly improved.

English was revised and improved in the present version.

Comments on the Quality of English Language

Minor editing of English language required

English was revised and improved in the present version.

We hope that our contribution is now ready to be accepted.

Reviewer 2 Report (Previous Reviewer 3)

Comments and Suggestions for Authors

Introduction: Please, add references from 2023 and 2024.

For the first paragraph only one reference was used. please, add some references related to this part. For second paragraph only seven references were used, related to the application and deposition methods. 

For the Introduction chapter, there are not enough references used, the used references are correlated to smaller part of the Introduction, and more recent papers (2023 and 2024) are missing. This was the major comment for  the previous version of the paper.

Materials and Methods: Three different quantities (5, 8 and 12 mL),

precursor’s volumes and concentrations: 5, 8, 12 ml at 0.1 M and 0.05 M. 

Why this parameters were chosen, using previous experience or literature data?

References : add references from 2023 and 2024.

Comments on the Quality of English Language

Minor editing of English language required

Author Response

We thank the reviewer for his time and suggestions for improving our manuscript. More references were added to the revised version of the manuscript. Detailed answers to the reviewer’s comments are provided bellow between lines. We must mention that there are still some formatting issues and perhaps minor English typos that will be solved in collaboration with mdpi editorial team after the manuscript acceptance. We hope that our contribution is now ready to be accepted.

Comments and Suggestions for Authors

Introduction: Please, add references from 2023 and 2024.

More newer references were added to Introduction in the revised improved version of the manuscript.

For the first paragraph only one reference was used. please, add some references related to this part. For second paragraph only seven references were used, related to the application and deposition methods. 

More newer references were added to the first paragraph revised improved version of the manuscript.

For the Introduction chapter, there are not enough references used, the used references are correlated to smaller part of the Introduction, and more recent papers (2023 and 2024) are missing. This was the major comment for  the previous version of the paper.

More newer references were added to Introduction in the revised improved version of the manuscript.

Materials and Methods: Three different quantities (5, 8 and 12 mL),

precursor’s volumes and concentrations: 5, 8, 12 ml at 0.1 M and 0.05 M. 

Why this parameters were chosen, using previous experience or literature data?

The parameters were chosen based on previous experience – to have a homogeneous deposition and to avoid excessive large agglomerations onto surface. Up to our knowdlege, there are no literature data regarding growth of such 3D structured pure WO3.

References : add references from 2023 and 2024.

More newer references were added to the revised improved version of the manuscript.

Comments on the Quality of English Language

Minor editing of English language required

English was revised and improved in the present version.

We hope that our contribution is now ready to be accepted.

Round 2

Reviewer 1 Report (Previous Reviewer 2)

Comments and Suggestions for Authors

Accept the manuscript in the present form.

Author Response

Thank you for your valuable help to improve our work.

Reviewer 2 Report (Previous Reviewer 3)

Comments and Suggestions for Authors

line 39 refrence is missing [2-?]

Comments on the Quality of English Language

 Minor editing of English language required

Author Response

Thank you for your valuable help to improve our work.

This manuscript is a resubmission of an earlier submission. The following is a list of the peer review reports and author responses from that submission.

Round 1

Reviewer 1 Report

Comments and Suggestions for Authors

The authors reported Raman spectra for different deposition conditions. The changes in the spectra are rather minor, and the results may depends on the deposition apparatus used. Thus I do not feel that the results are interesting in terms of general material science.

They noted

“Comparing the Raman spectra (Figure 3) obtained for WO3 films made by spray deposition method with results reported in the literature using other deposition techniques (Chemical Vapor Deposition, Chemical Solution Deposition, or Dip-Coating), one can observe a dominant presence of vibration modes at 270 cm-1, 714 cm-1, and 805 cm-1.”

but the references are not given, and one cannot see how are the results reported in the literature. The present results will be meaningful only if the difference from the literatures is significant and if the difference is successfully interpreted.

“Deposition of WO3 by spray-coating induces slight shifts in the frequency of the Raman bands (Figure 4c γ (W-O)). These shifts may be caused by changes in the local environment of the tungsten atoms or alterations in the interactions between atoms and light.”

The discussion is too vague, seems useless.

“All these changes are induced by deviations from the ideal stoichiometry of the WO3 composition, resulting in a variation of the atomic ratio between tungsten and oxygen”

To claim that, elemental composition should be measured by some chemical analysis.

“The disappearance of symmetric νs (O−W−O) may be caused by chemical reactions leading to modification or destruction of terminal oxygen groups in the WO3 lattice. This could result from interactions with compounds during the deposition process.”

I did not understand the discussion. (What is “terminal oxygen group” ? What is “compounds” ?)

Actually, it is not easy to interpret change in Raman signals. Crystallinity had better be discussed based on XRD. To check stoichiometry, chemical analyses, e.g., XPS, will be better. Devices should be fabricated and characterized to discus effects on device performance. They just measured Raman, and failed to derive information useful for material scientists. Thus this paper should be rejected.

Comments on the Quality of English Language

no big problem

Author Response

We thank the reviewer for time and effort but we are unpleasantly surprised by his superficial approach and subjectivity. As one can see, the reviewer provided only a personal opinion and a few comments regarding a small part of the manuscript without any attempt of suggesting improvements or a critical scientific perspective on the presented results and in total disagreement with the other two reviewers. With all due respect, we choose not to comment more on the reviewer observations.

Reviewer 2 Report

Comments and Suggestions for Authors

The manuscript under review is an interesting study on a technology-relevant topic – development of WO3 thin films for sensors applications and its detailed Raman spectroscopy study. The method of synthesis is inexpensive and flexible. The manuscript is well-organized and illustrated. It is of high relevance to the audience of the journal Nanomaterials. However, it needs to be improved before acceptance for publication following the issues below.

1.    Rewrite the Abstract adding some values and combining the first 2 sentences.

2.    Replace “growing thickness” with “by increasing the thickness”.

3.    Shorten the Introduction part, it is extremely long. Briefly mention the different fields of applications of WO3 films such as photochromic [Refs. from your list] electrochromic [C.G.Granqvist, Handbook of inorganic electrochromic materials;], and gasochromic […] “smart windows”.

4.    Briefly describe the methods for WO3 deposition such as PVD (thermal evaporation [https://doi.org/10.1016/j.matlet.2017.02.078; https://doi.org/10.1016/S0925-4005(99)00504-3], DC- [https://doi.org/10.1016/S0925-4005(99)00360-3], RF- [https://doi.org/10.1016/S0925-4005(99)00504-3] and magnetron sputtering [https://doi.org/10.1063/1.4880162]) and CVD [https://doi.org/10.1051/epjap:2000159], sol-gel [https://doi.org/10.1016/S0925-4005(99)00504-3]. Underline the cost-effectiveness of your non-vacuum method. What is new in the synthesis?

5.    Change “device” with “equipment” in Materials and Methods.

6.    Why do you anneal the samples if you do not have organic precursors? To increase the thin films crystallinity? How did you select the annealing temperature? Add the optimization data if any.

7.    Improve the contrast-brightness ratio of Fig. 1.

8.    In all Raman spectra give the peaks assignments from Table 1 not the peaks positions. Can you add some Raman spectra of the as-deposited samples and annealed at lower temperatures to see the Raman peaks evolution with temperature?

9.    Give the orientations of the most intense peaks in the XRD spectra.

10.                   Explain better the bands between 270 and 700 cm-1 and Ref. [19].

11.                   Use the standard journal abbreviations in the References.

12.                   It would be nice if you could add some sensor-related data for the newly developed thin films intended for sensors applications.

Author Response

We thank the reviewer for his time and valuable comments and advice that leads to a improved version of our paper. We considered and accommodated all recommendations and we hope that now the manuscript is ready for publication.  Please find bellow detailed answers to the reviewer’s comments.

  1. Rewrite the Abstract adding some values and combining the first 2 sentences.

The abstract was rephrased.

  1. Replace “growing thickness” with “by increasing the thickness”.

Done

  1. Shorten the Introduction part, it is extremely long. Briefly mention the different fields of applications of WO3 films such as photochromic [Refs. from your list] electrochromic [C.G.Granqvist, Handbook of inorganic electrochromic materials;], and gasochromic […] “smart windows”.

The information was added to the manuscript.

  1. Briefly describe the methods for WO3 deposition such as PVD (thermal evaporation [https://doi.org/10.1016/j.matlet.2017.02.078; https://doi.org/10.1016/S0925-4005(99)00504-3], DC- [https://doi.org/10.1016/S0925-4005(99)00360-3], RF- [https://doi.org/10.1016/S0925-4005(99)00504-3] and magnetron sputtering [https://doi.org/10.1063/1.4880162]) and CVD [https://doi.org/10.1051/epjap:2000159], sol-gel [https://doi.org/10.1016/S0925-4005(99)00504-3]. Underline the cost-effectiveness of your non-vacuum method. What is new in the synthesis?

The information was added to the manuscript. Information was added about the advantages of the method and novelty.

  1. Change “device” with “equipment” in Materials and Methods.

Done

  1. Why do you anneal the samples if you do not have organic precursors? To increase the thin films crystallinity? How did you select the annealing temperature? Add the optimization data if any.

The reason for annealing was improving the adhesion to the substrate and slightly improved crystallinity. The specific annealing conditions were chosen based on our previous metal oxides growth experience.

  1. Improve the contrast-brightness ratio of Fig. 1.

Done.

  1. In all Raman spectra give the peaks assignments from Table 1 not the peaks positions. Can you add some Raman spectra of the as-deposited samples and annealed at lower temperatures to see the Raman peaks evolution with temperature?

With all due respect, we prefer to keep the mark on the position in the figure. Adding the peaks assignments according to the table would overload the graphs.

The annealing does not induce significant changes in the Raman spectra since the precursor we use is tungsten peroxide but only a slight increase in main peaks intensity. It will make no difference including an as deposited spectrum but just load the manuscript. We hope that the reviewer would agree with us on this point.

  1. Give the orientations of the most intense peaks in the XRD spectra.

The orientations are mentioned in figure 2.

  1. Explain better the bands between 270 and 700 cm-1 and Ref. [19].

The bands between 270-700 are now better explained in the manuscript. The reference 19 was wrongly cited in the previous version. The improved manuscript include the correct references.

  1. Use the standard journal abbreviations in the References.

Done. Please note that format and text corrections would be fixed also during the proofing together with mdpi team.

  1. It would be nice if you could add some sensor-related data for the newly developed thin films intended for sensors applications.

We would prefer to keep this communication limited to its subject. Sensor’s fabrication and behavior will be the subject of a further communication when the work and specific data would be experimentally validated. We hope that the reviewer understands.

Reviewer 3 Report

Comments and Suggestions for Authors

Introduction: it is very unusuall to have Introduction chapter with only 4  refrences. Please, include appropriate references.

XRD

sentence is not correlated to the cited refrences.  Please add some comments on refrences used.

In addition, multiple reports related to the WO3 films and nanostructures exhibit monoclinic crystal structures [8; 9; 10].

Figure 3. Raman spectroscopy results for WO3 thin films grown at precursor’s volumes and concentrations: 5, 8, 12 ml at a 0.05 M and b 0.1 M.

Above this picture there is (b). Is is (a) missing, or b is and error?

The bands between 270 and 700 cm−1 are the typical modes indicating the crystalline quality of the WO3 films [19].  What is relation with the refrence cited?

Technical remark: pease use the same font in paper.

Reference list used have to be incorporeted in the paper in much better way, to be clear why the refernces were used, and purpose and which relevant results are important for using.

Reviwer have the impresion that refrence list is in the paper only for reason that this is part of the required instruction for authors.

Comments on the Quality of English Language

Minor editing of English language required.

Author Response

We thank the reviewer for his time and valuable comments and advice that leads to a improved version of our paper. We considered and accommodated all recommendations and we hope that now the manuscript is ready for publication.  Please find bellow detailed answers to the reviewer’s comments.

Introduction: it is very unusuall to have Introduction chapter with only 4  refrences. Please, include appropriate references.

 Thank you for your observation. The introduction was improved in the new version of the manuscript.

XRD

sentence is not correlated to the cited refrences.  Please add some comments on refrences used.

In addition, multiple reports related to the WO3 films and nanostructures exhibit monoclinic crystal structures [8; 9; 10].

The XRD part was improved in the revised version of the manuscript. 

Figure 3. Raman spectroscopy results for WO3 thin films grown at precursor’s volumes and concentrations: 5, 8, 12 ml at a 0.05 M and b 0.1 M.

Above this picture there is (b). Is is (a) missing, or b is and error?

Thank you for your observation, b was an error. The figure was corrected.

The bands between 270 and 700 cm−1 are the typical modes indicating the crystalline quality of the WO3 films [19].  What is relation with the refrence cited?

Thank you for your observation. Reorganizing the manuscript, some errors regarding references were committed. By mistake, the papers corresponding to 19 and 20 in a previous version were partially reassigned/removed. The correct references are now references 20 and 21.

Technical remark: pease use the same font in paper.

 Thank you for the observation. There were some problems with the journal template incompatibility with our MS Office versions. Any remining problems will be fixed during the editorial process after manuscript acceptation for publication. 

Reference list used have to be incorporeted in the paper in much better way, to be clear why the refernces were used, and purpose and which relevant results are important for using.

Reviwer have the impresion that refrence list is in the paper only for reason that this is part of the required instruction for authors.

References were better included in the text. We hope that the present version is acceptable for publication.